# Crowdsourced biodiversity monitoring fills gaps in global plant trait mapping

Plant functional traits are fundamental to ecosystem dynamics and Earth system processes, but their global characterization is limited by available field surveys and trait measurements. Recent expansions in biodiversity data aggregation—including vegetation surveys, citizen science observations, and trait measurements—offer new opportunities to overcome these constraints. Here we demonstrate that combining these diverse data sources with high-resolution Earth observation data enables accurate modeling of key plant traits at up to 1 km$^2$ resolution. Our approach achieves correlations up to 0.63 (15 of 31 traits exceeding 0.50) and improved spatial transferability, effectively bridging gaps in under-sampled regions. By capturing a broad range of traits with high spatial coverage, these maps can enhance understanding of plant community properties and ecosystem functioning, while serving as tools for modeling global biogeochemical processes and informing conservation efforts. Our framework highlights the power of crowdsourced biodiversity data in addressing longstanding extrapolation challenges in global plant trait modeling, with continued advancements in data collection and remote sensing poised to further refine trait-based understanding of the biosphere.

The diversity and distribution of plants shape ecosystem function and influence global biogeochemical cycles[1–3]. Since Humboldt's early explorations of plant geography in 1807, ecologists have sought to understand how plant properties govern species' success across environmental gradients and their role in Earth system processes[4–7]. Despite significant advances, a comprehensive global understanding of plant functional traits and their implications for ecosystem resilience remains incomplete[8,9].

Plant functional traits—measurable characteristics influencing growth, reproduction, and survival—are central to ecological strategies and ecosystem modeling[10–13]. Traits such as leaf area, wood density, and seed mass reflect trade-offs in resource use, stress tolerance, and competition, shaping plant community assembly[14]. Integrating these traits into global vegetation and Earth system models is crucial for refining projections of energy, carbon, and water cycles[15,16]. However, producing spatially explicit, high-resolution trait maps requires extensive in situ trait measurements—a challenge that remains largely unmet.

Earth observation technologies provide valuable insights into vegetation properties, offering continuous, high-resolution data on surface reflectance, vegetation structure, climate, and soil conditions[17–22]. Yet, their utility for plant trait modeling remains constrained by limited in situ trait observations for calibration and validation. While in situ trait measurements are accurate, they are geographically sparse and labor-intensive[23]. Global initiatives such as TRY and BIEN have compiled extensive trait databases[24–26], but they often lack the spatial coverage necessary for coherently mapping plant functional composition at fine scales. Consequently, satellite-based trait extrapolations carry significant uncertainties[27,28].

Crowdsourced ecological data offer a promising avenue to address these gaps. Expert-led vegetation surveys, such as those aggregated by the sPlot database, document species co-occurrence and abundance around the planet[29–31]. However, many regions remain underrepresented (Fig. 1, SCI). Meanwhile, crowdsourced biodiversity databases like the Global Biodiversity Information Facility (GBIF), which aggregates records from citizen science platforms such as

✉ e-mail: daniel.lusk@geosense.uni-freiburg.de

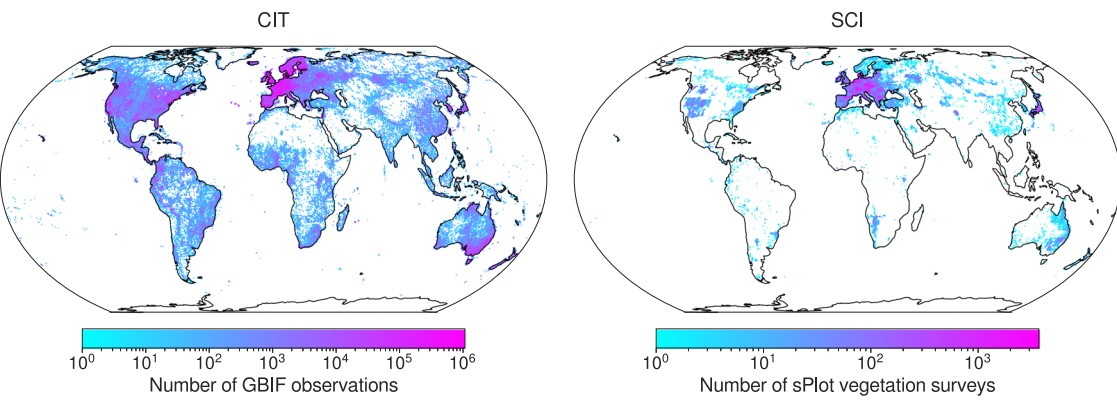

**Fig. 1 | Density maps of citizen science observations (CIT) and vegetation surveys (SCI) after filtering.** Filtering included matching with trait data from the TRY trait database, spatial aggregation, and minimum and maximum abundance filtering. Density is in number of observations (CIT) and surveys (SCI) per 1 km² and aggregated to 55 km² for visualization.

iNaturalist and Pl@ntNet alongside many other sources, have contributed over half a billion plant observations (Fig. 1, CIT), vastly expanding global species distribution datasets[32–34]. Though these data exhibit spatial and taxonomic biases, they hold great potential for plant trait inference. Recent work by Wolf et al.[28] showed that combining citizen science occurrences with trait databases can yield community-weighted mean trait estimates closely aligned with those from vegetation plot data.

Each data source—trait measurements, vegetation surveys, and citizen science observations—has strengths and limitations. Here, we present a data integration framework that combines these sources with Earth observation to generate continuous, high-resolution (up to 1 km²) global maps of 31 key plant traits. Our approach integrates environmental data on surface reflectance, climate, soil properties, and vegetation structure with species observations from GBIF, expert vegetation surveys from sPlot, and trait records from TRY to improve spatial coverage and accuracy while quantifying uncertainty and spatial transferability (Fig. 2).

In this work, we predict community-weighted mean trait values globally using three data subsets: (1) scientific vegetation surveys alone (SCI), (2) citizen science observations alone (CIT), and (3) a combined approach (COMB). Using spatial cross-validation with independent vegetation survey data, we assess model performance and geographic generalizability across scales from 1 to 222 km². This multi-resolution modeling strategy allows us to explore resolution-performance trade-offs and identify the optimal spatial scales for specific traits. Our approach achieves unprecedented accuracy for many traits and, to our knowledge, maps several traits at high resolution for the first time. By integrating diverse data sources, we produce the most precise large-scale trait maps to date, significantly improving spatial coverage and predictive reliability. These advances provide a robust foundation for refining ecosystem models and predicting global vegetation dynamics with greater confidence.

## Results

### Density and extent of citizen science and survey data
The three trait data subsets—vegetation surveys (SCI), citizen science observations (CIT), and their combination (COMB)—exhibited distinct patterns of observation density and geographic coverage. Prior to matching with trait measurements from the TRY trait database, CIT contained 339,971,350 observations of 314,217 species, while SCI consisted of 2,534,183 plots recording 52,942,365 observations of 91,603 species. After matching with TRY, CIT retained 89% of observations ($n$ = 303,288,097) and 28% of species ($n$ = 88,802), while SCI retained 84% of plot-level relative abundance ($n$ = 45,037,375

observations) and 43% of species ($n$ = 39,286). Following spatial aggregation for matching with environmental predictors, CIT (and therefore COMB) displayed markedly greater spatial coverage than SCI alone, though both subsets showed spatial clustering in Europe, North America, Japan, and Australia (Fig. 1).

### Global trait maps and spatial transferability
We modeled the global distributions of 31 plant traits (Table 1) across five spatial resolutions: 1 km² (-0.01°), 22 km² (-0.2°), 55 km² (-0.5°), 111 km² (-1°), and 222 km² (-2°) (Fig. 3). To account for differences in observation density and geographic coverage, models were generated for each of the three trait data subsets: SCI, CIT, and COMB. We used gradient-boosted decision trees, which are well-suited for capturing complex, nonlinear relationships, with environmental predictors aggregated at each corresponding spatial resolution[35]. Rather than deriving coarser-resolution maps from higher-resolution outputs, we trained separate models at each scale to prevent biases from simple upscaling. This approach yielded a total of 465 models (31 traits × 5 resolutions × 3 trait data subsets).

Final trait maps, along with each map's corresponding coefficient of variation (COV) and area of applicability (AOA) masks, were generated as GeoTIFF rasters. COMB maps at 1 km² resolution can be viewed online via an *interactive map viewer* (https://global-traits.projects.earthengine.app/view/global-traits) or downloaded directly (see "Data availability"). Predictions were made for all pixels where sufficient predictor data were present with the exclusion of permanent water bodies. In the interest of transparency and reproducibility for future work, all trait maps are made available. Given the range of model performance, map users are encouraged to carefully consider their specific use cases alongside the provided model performance and uncertainty indicators (COV and AOA). Plant functional type-specific maps are also available for download. Citizen science observations, with their significantly greater geographic extent and sample size, have the potential to enhance vegetation survey data by allowing models to predict with greater confidence in data-deficient regions. To evaluate this, we calculated two metrics for each trait map and model during spatial cross-validation: area of applicability[36] and coefficient of variation (see "Methods") (Fig. 4). For this portion of the analysis, 1 km² maps were primarily considered to ensure comparison at the finest grain size available, though area of applicability was also compared across resolutions.

The applicability of models to new domains, or their AOA, is determined by how similar the new region's predictors are to those used during the model's training[36]. Incorporating citizen science data alongside vegetation surveys increased the AOA for all traits at all

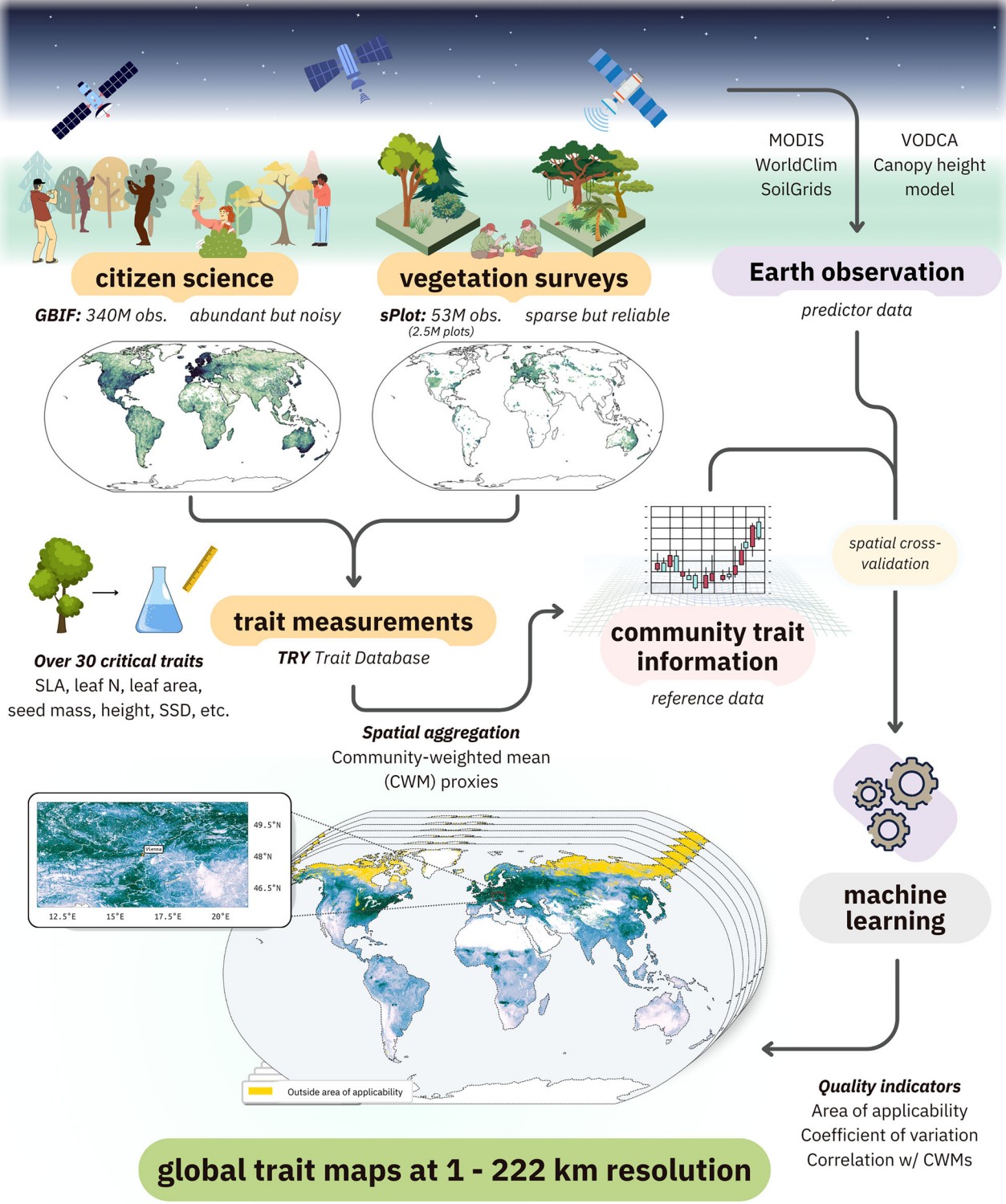

**Fig. 2 | Workflow for modeling plant functional traits using citizen science observations, vegetation surveys, trait measurement data, and Earth observation.** The approach integrates community composition and trait information as reference data to train machine learning models, which generate global community-weighted mean trait maps at up to 1 km² resolution.

resolutions, with an average gain of 2.43 percentage points. The largest improvement was 9.22 for wood fiber length at 1 km² resolution, while the smallest was 0.04 for leaf width at 222 km² resolution. In particular, trait maps at finer resolutions benefited more on average from the inclusion of citizen science trait data, while the benefits became smaller at coarser resolutions (Fig. 4d).

The coefficient of variation (COV) is another indicator of model uncertainty derived by comparing the predictions made by each model during cross-validation. For all models, the COV was consistently lower on average for COMB maps than for SCI maps. In particular, COMB maps showed better generalization across all biomes, including regions where observations are especially sparse for both

**Table 1 | Plant functional traits used for modeling and trait map generation**

| Trait | TRY trait name | Unit |
|---|---|---|
| Conduit element length | Wood vessel element length; stem conduit (vessel and tracheids) element length | μm |
| Dispersal unit length | Dispersal unit length | mm |
| LDMC | Leaf dry mass per leaf fresh mass (leaf dry matter content, LDMC) | $g\,g^{-1}$ |
| Leaf area | Leaf area (in case of compound leaves: leaflet, undefined if petiole is in- or excluded) | $mm^2$ |
| Leaf C | Leaf carbon (C) content per leaf dry mass | $mg\,g^{-1}$ |
| Leaf C/N ratio | Leaf carbon/nitrogen (C/N) ratio | $g\,g^{-1}$ |
| Leaf delta 15N | Leaf nitrogen (N) isotope signature (delta 15N) | ppm |
| Leaf dry mass | Leaf dry mass (single leaf) | g |
| Leaf fresh mass | Leaf fresh mass | g |
| Leaf length | Leaf length | mm |
| Leaf N (area) | Leaf nitrogen (N) content per leaf area | $g\,m^{-2}$ |
| Leaf N (mass) | Leaf nitrogen (N) content per leaf dry mass | $mg\,g^{-1}$ |
| Leaf P | Leaf phosphorus (P) content per leaf dry mass | $mg\,g^{-1}$ |
| Leaf thickness | Leaf thickness | mm |
| Leaf water content | Leaf water content per leaf dry mass (not saturated) | $g\,g^{-1}$ |
| Leaf width | Leaf width | mm |
| Plant height | Plant height (vegetative) | m |
| Rooting depth | Root rooting depth | m |
| Seed germination rate | Seed germination rate (germination efficiency) | days |
| Seed length | Seed length | mm |
| Seed mass | Seed dry mass | mg |
| Seed number | Seed number per reproduction unit | - |
| SLA | Leaf area per leaf dry mass (specific leaf area, SLA or 1/LMA): undefined if petiole is in- or excluded) | $m^2\,kg^{-1}$ |
| SRL | Root length per root dry mass (specific root length, SRL) | $cm\,g^{-1}$ |
| SRL (fine) | Fine root length per fine root dry mass (specific fine root length, SRL) | $cm\,g^{-1}$ |
| SSD | Stem specific density (SSD) or wood density (stem dry mass per stem fresh volume) | $g\,cm^{-3}$ |
| Stem conduit density | Stem conduit density (vessels and tracheids) | $mm^{-2}$ |
| Stem conduit diameter | Stem conduit diameter (vessels, tracheids) | μm |
| Stem diameter | Stem diameter | m |
| Wood fiber lengths | Wood fiber lengths | μm |
| Wood ray density | Wood rays per millimeter (wood ray density) | $mm^{-1}$ |

"Trait" refers to a shortened version of the "TRY trait name". "TRY trait name" refers to the name used in the TRY trait database.

citizen science observations and vegetation surveys, such as desert, alpine, and wetland regions.

**Model performance across trait data subsets at 1 km² resolution**

To assess predictive accuracy at high spatial resolution, we compared model performance across the three trait data subsets (SCI, CIT, and COMB) at 1 km² resolution, using spatial cross-validation with independent vegetation survey community-weighted mean (CWM) trait data not employed in model training (Fig. 5a–c). Model performance for all traits across all trait data subsets and resolutions can be found in Table S2.

SCI and COMB models consistently outperformed CIT models in terms of correlation coefficient ($r$) and normalized root mean squared error (nRMSE). COMB models maintained similar $r$ and nRMSE values to those trained solely on SCI data, with an average increase in $r$ by 0.12 compared to CIT models. SCI models marginally outperformed COMB models on average, but COMB models consistently demonstrated lower mean coefficients of variation (Fig. 4a, b and Table S2). At 1 km² resolution, SCI models had $r \geq 0.5$ for 16 out of 31 traits, COMB models for 15 traits, and CIT models for 6 traits. The traits with the highest predictive accuracy across both COMB and SCI models included leaf N (area), specific leaf area (SLA), rooting depth, stem specific density (SSD), stem conduit diameter, and leaf area. Model prediction error (nRMSE) across biomes (Fig. 5c) was generally lowest for SCI models and only slightly higher for COMB models, while CIT models consistently exhibited higher error across most biome types. Wetland regions displayed a wide range in error for all models. Notably, COMB models displayed robust predictive accuracy across biomes, often outperforming SCI models slightly in data-sparse regions (Fig. S3).

**Map accuracy of combined trait data across spatial scales**

To evaluate how spatial resolution affected map accuracy, we compared Pearson's $r$ for COMB models across five resolutions (1–222 km²), where training data were aggregated at each resolution prior to model training to avoid simple upscaling effects (Fig. 5d). Accuracy generally improved with increasing spatial aggregation, with most traits (18 of 31) showing peak performance at 22 km² or 55 km² resolution. Beyond this point, accuracy tended to plateau or slightly decline, suggesting diminishing predictability at coarser scales for many traits. Similar trends were observed for SCI and CIT models (Table S.2).

**Comparison with existing trait map products**

To benchmark model performance, we compared COMB and CIT predictions with existing global trait maps generated through model-based extrapolations from sparse trait data, though using differing methodologies (Table 2). Three widely studied traits—specific leaf area (SLA), leaf N (mass), and leaf N (area)—were selected due to their broad overlap with prior trait mapping efforts.

At all resolutions and across all three traits, COMB predictions from this study exhibited stronger correspondence with independent, held-out vegetation survey data than previous trait maps. COMB models consistently produced the highest Pearson correlation coefficients ($r$), with values as high as 0.68 at 22 km² resolution for leaf N (area). Even at coarser scales, COMB models retained high predictive accuracy, with average $r$ values exceeding those of earlier studies. Notably, citizen science-only models (CIT), which employed no vegetation survey data during training, also demonstrated competitive performance, often outperforming previously published trait maps and achieving the second-highest correlations for most trait-resolution combinations.

**Importance of environmental predictors**

We assessed the relative importance of each Earth observation predictor to determine which predictor datasets contributed most to model performance (Fig. S5). The predictor datasets included World-Clim bioclimatic variables ($n = 6$), MODIS land surface reflectance features ($n = 72$), SoilGrids soil properties ($n = 61$), VODCA vegetation optical depth features ($n = 9$), and global canopy height data ($n = 2$)[19–22,37].

For models trained with the combined dataset (COMB), MODIS surface reflectance emerged as the most influential predictor set overall (mean importance = 0.083), followed by WorldClim bioclimatic variables (0.051), SoilGrids soil properties (0.042), canopy height (0.008), and vegetation optical depth (0.006). Although MODIS, WorldClim, and SoilGrids contributed more strongly to trait

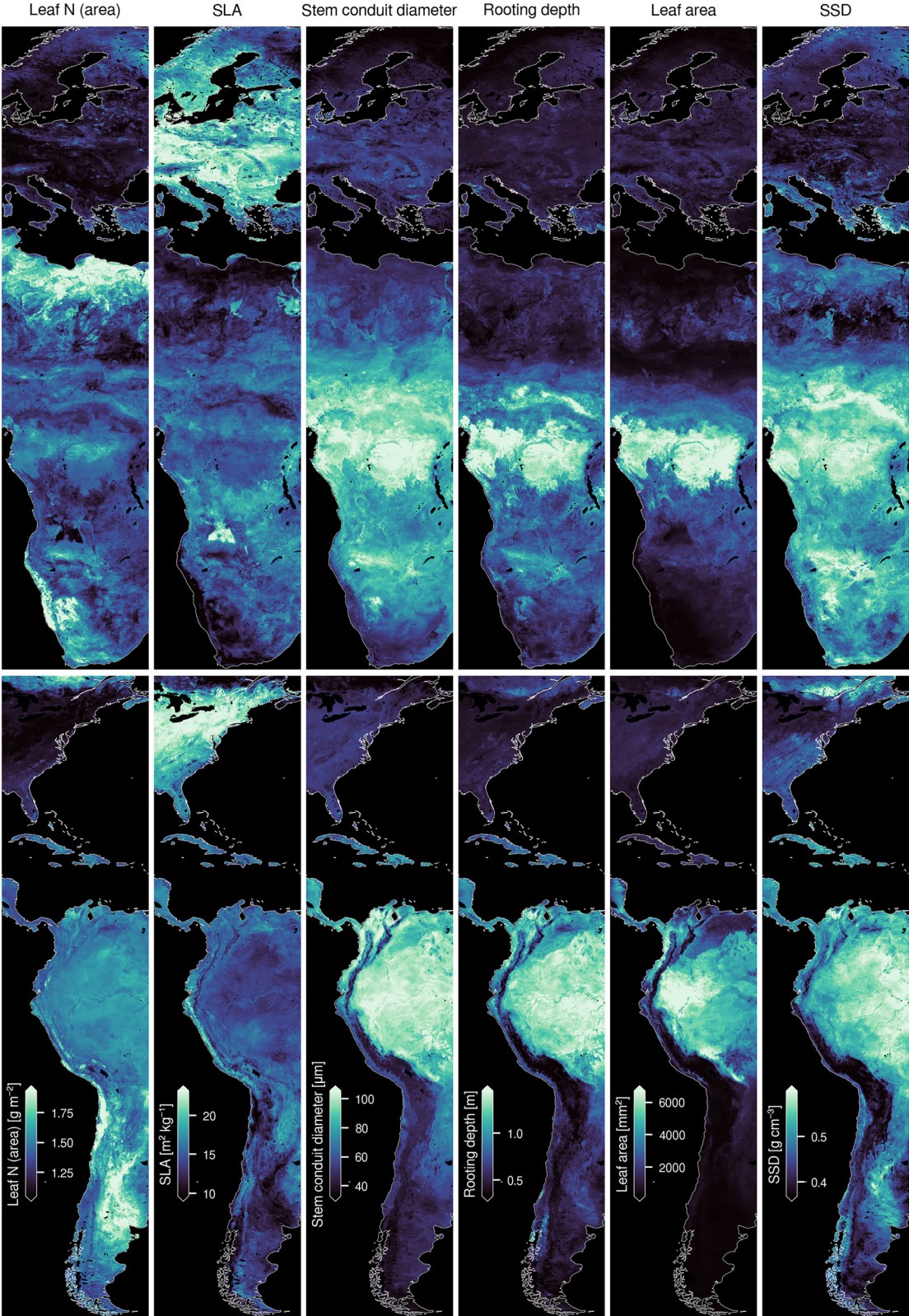

**Fig. 3 | Trait maps based on combined citizen science and vegetation survey data (COMB) at 1 km² resolution.** Here, maps are visualized across two longitudinal sections of Earth: northern Europe to southern Africa (top row) and eastern North America to western South America (bottom row) and are in WGS84 projection (EPSG:4326). Equal area trait maps with full global extent can be *viewed online* (https://global-traits.projects.earthengine.app/view/global-traits) or downloaded directly (see "Data availability")[106].

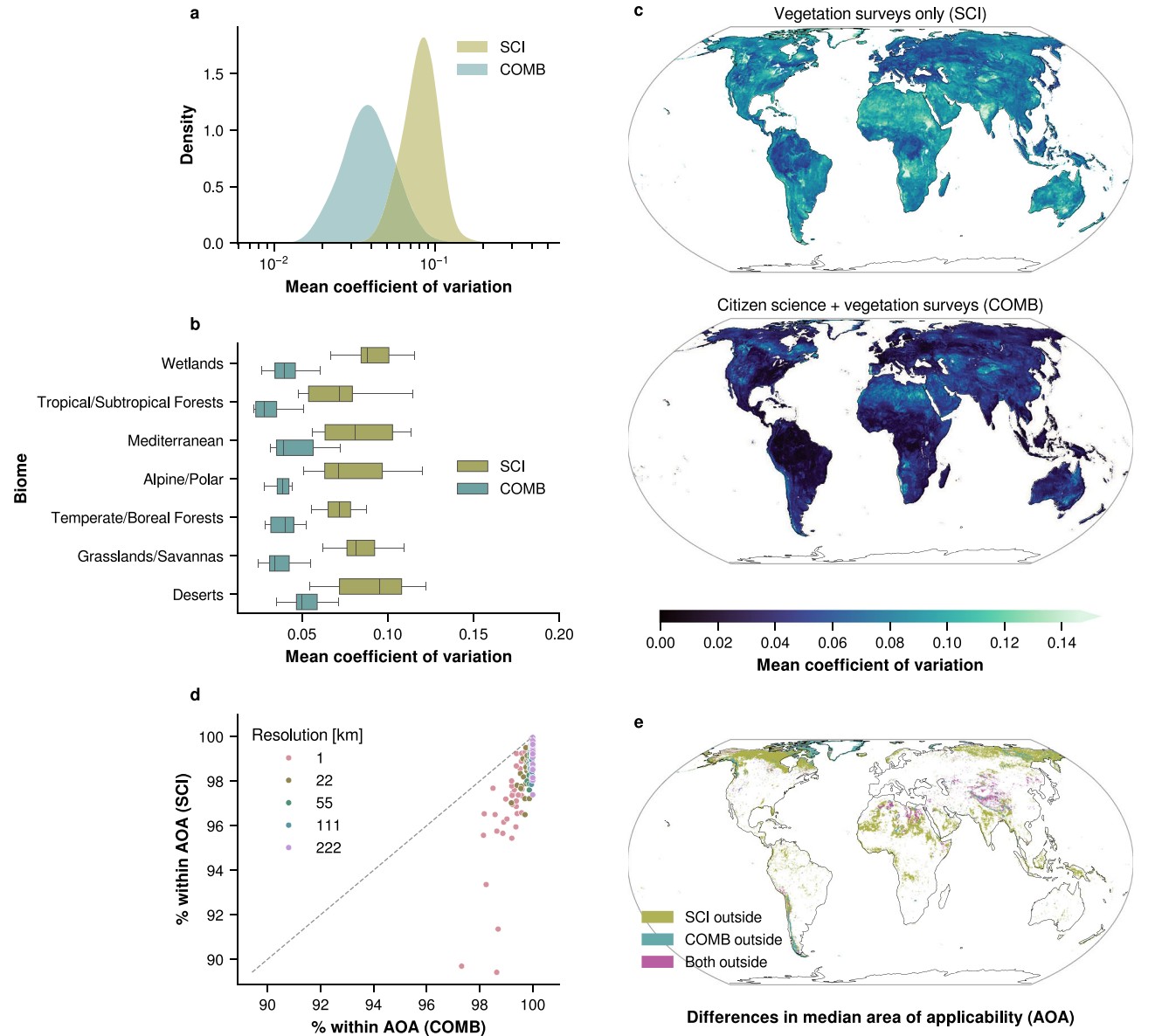

**Fig. 4 | Spatial transferability metrics for traits with *r* ≥ 0.5 at 1 km² resolution.** **a** Mean coefficient of variation (COV) from spatial cross-validation for models using only vegetation survey trait data (SCI) and those combining vegetation survey and citizen science data (COMB). **b** Mean COV for SCI and COMB models (*n* = 16 trait models) by biome. Boxes indicate the 25th to 75th percentiles, with whiskers showing the interquartile range. See Table S4 for more information on biome definition. **c** Pixel-wise mean COV for SCI (top) and COMB (bottom). Including citizen science data generally lowered the pixel-wise COV compared to using vegetation survey data alone. **d** Scatter plot of the area of applicability (AOA) for SCI vs. COMB models across different resolutions. The percentage of prediction pixels within the AOA is shown, with colors representing model resolution. A one-to-one line is included for reference. COMB models consistently showed a larger AOA than SCI across all resolutions. **e** Pixel-wise median AOA for SCI and COMB. Olive areas indicate where only SCI predictions are unreliable, teal where only COMB predictions are unreliable, and magenta where both are outside the AOA.

predictions on average, the importance of individual predictors varied widely across traits. Notably, the highest single predictor importance was observed for a WorldClim variable (0.139, stem conduit density), followed by MODIS (0.091, specific leaf area), SoilGrids (0.025, rooting depth), canopy height (0.023, plant height), and vegetation optical depth (0.013, leaf N [mass]).

## Discussion

Integrating structured vegetation surveys and opportunistic citizen science observations provides a means to expand spatial coverage, reduce extrapolation error, and improve trait predictions at a global scale. While citizen science data inherently contains noise and bias due to unstructured and opportunistic sampling[38–41], its broad geographic

distribution and taxonomic coverage complement the systematic yet spatially limited nature of expert vegetation surveys. Models trained on a combination of these data sources (COMB) achieved predictive power comparable to those trained solely on vegetation surveys (SCI; Fig. 5), while also benefiting from markedly improved spatial transferability (Fig. 4).

Predictive performance was similar between SCI and COMB models, likely due to the down-weighting of citizen science samples in the COMB subset aimed at encouraging the models to prioritize the more structured and reliable patterns from vegetation surveys. In fact, when examined across biomes, COMB models tended to moderately outperform SCI models (Fig. S3). Because $R^2$ can be biased by differences in trait variance across biomes, however, we emphasize nRMSE

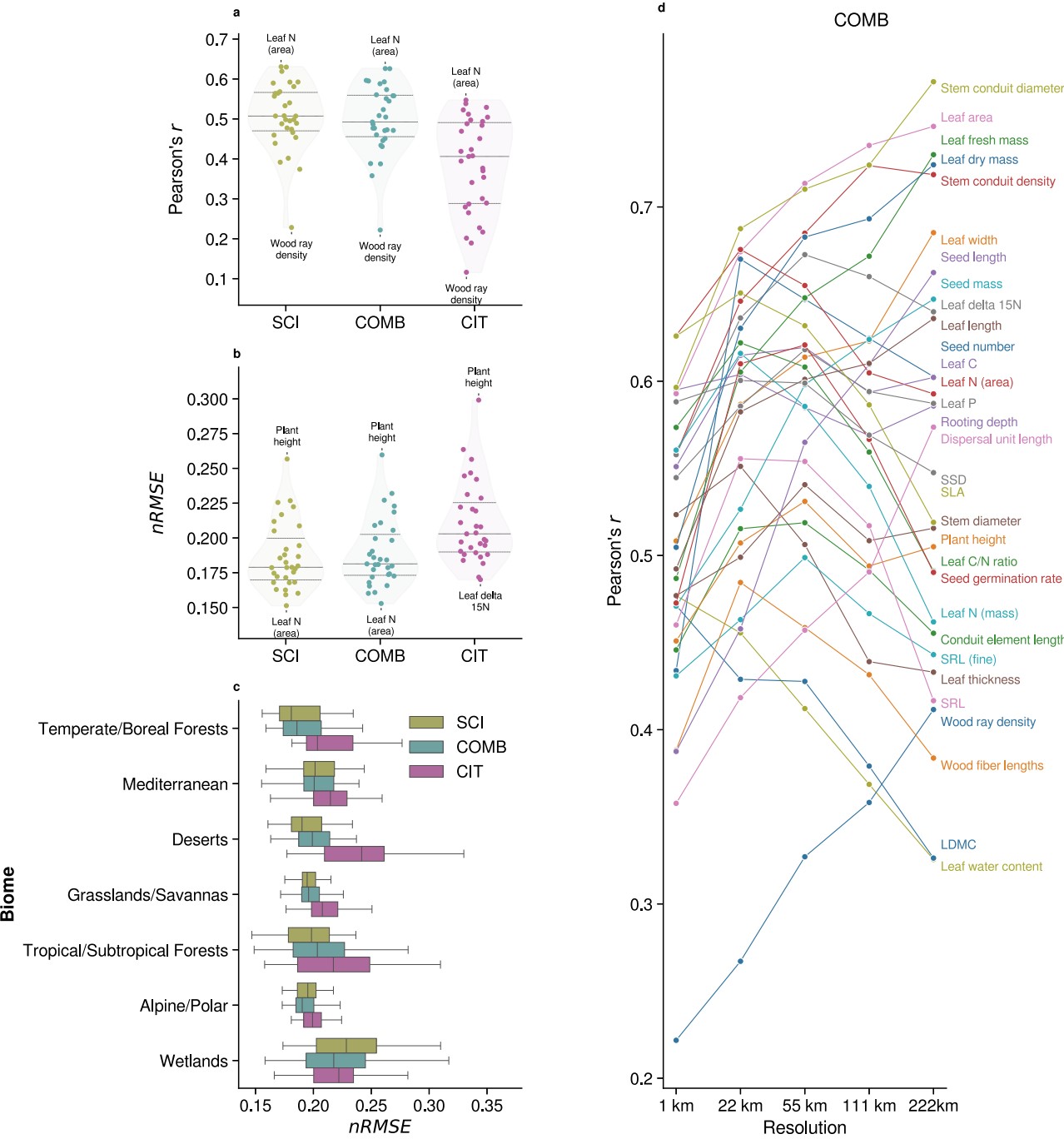

**Fig. 5 | Model performance across trait data subsets.** Trait data subsets include vegetation survey data alone (SCI), citizen science data alone (CIT), and a combination of the two (COMB). All metrics are the result of validation against spatially independent, held-out vegetation survey community-weighted mean trait values. Model performance for all traits across all trait data subsets and resolutions can be found in Table S2. Pearson correlation coefficient *r* (**a**) and normalized root mean squared error *nRMSE* (**b**) of trait models for SCI, COMB, and CIT trait data subsets. The points indicate individual trait model performance, while the encompassing violin plots represent the performance distribution for each trait subset. Inner horizontal bars represent the 75th, 50th (median), and 25th quantiles. **c** *nRMSE* of SCI, COMB, and CIT models (*n* = 31 trait models) across biomes. Boxes indicate the 25th to 75th percentiles, with whiskers showing the interquartile range. See Fig. S3 for Pearson's *r* and *R*² by biome, as well as Table S4 for more information about biome selection. **d**. Pearson's correlation coefficient *r* for COMB models trained on data aggregated at five spatial resolutions. For fold-wise mean and variance, see Fig. S2.

as a more reliable indicator of comparative model error in these cases. The inclusion of citizen science observations also resulted in a consistent reduction in the coefficient of variation and an expanded area of applicability on average and across biomes. This reduction in uncertainty indicates that, while vegetation surveys offer higher-quality trait data, citizen science observations can enhance spatial generalizability and predictive confidence, particularly in regions with limited survey coverage. These benefits were especially pronounced in remote and under-sampled regions, such as alpine/boreal, tropical, and wetland zones, where sparse vegetation surveys limit trait mapping efforts[25,29]. Overall, sampling effort significantly influenced model performance across biomes, with COMB models exhibiting reduced

**Table 2 | Comparison of trait map performance with existing products**

| Author | Resolution (km) | SLA | Leaf N (mass) | Leaf N (area) |
|---|---|---|---|---|
| This study (COMB) | 1 | **0.63** | **0.56** | **0.63** |
| | 22 | **0.65** | **0.62** | **0.68** |
| | 55 | **0.63** | **0.59** | **0.65** |
| | 111 | **0.59** | **0.54** | **0.60** |
| | 222 | **0.52** | **0.46** | **0.59** |
| This study (CIT) | 1 | *0.53* | *0.49* | *0.55* |
| | 22 | *0.45* | *0.49* | *0.53* |
| | 55 | 0.44 | *0.47* | *0.54* |
| | 111 | *0.42* | *0.44* | *0.52* |
| | 222 | 0.41 | *0.37* | *0.51* |
| van Bodegom et al.[83] | 55 | 0.33 | - | - |
| | 111 | 0.23 | - | - |
| | 222 | 0.24 | - | - |
| Boonman et al.[80] | 55 | 0.43 | 0.11 | 0.44 |
| | 111 | 0.37 | 0.08 | 0.42 |
| | 222 | 0.37 | 0.12 | 0.41 |
| Butler et al.[54] | 55 | 0.29 | 0.20 | 0.39 |
| | 111 | 0.36 | 0.29 | 0.37 |
| | 222 | 0.29 | 0.25 | 0.33 |
| Madani et al.[81] | 55 | 0.10 | - | - |
| | 111 | 0.25 | - | - |
| | 222 | 0.25 | - | - |
| Moreno et al.[45] | 1 | 0.38 | 0.26 | - |
| | 22 | 0.38 | 0.31 | - |
| | 55 | 0.38 | 0.12 | - |
| | 111 | 0.40 | 0.21 | - |
| | 222 | *0.44* | 0.21 | - |
| Schiller et al.[49] | 55 | *0.47* | 0.38 | 0.50 |
| | 111 | 0.38 | 0.35 | 0.46 |
| | 222 | 0.39 | 0.31 | 0.49 |
| Vallicrosa et al.[82] | 1 | - | 0.29 | - |
| | 22 | - | 0.37 | - |
| | 55 | - | 0.20 | - |
| | 111 | - | 0.33 | - |
| | 222 | - | 0.30 | - |
| Wolf et al.[28] | 22 | 0.41 | 0.31 | 0.38 |
| | 55 | 0.42 | 0.27 | 0.41 |
| | 111 | 0.32 | 0.29 | 0.34 |
| | 222 | 0.31 | 0.36 | 0.37 |

Pearson's correlation coefficient *r* of each trait map in relation to spatially-independent gridded sPlot community-weighted means (CWM) unused in model training at different resolutions (1–222 km²). Bold values represent the highest correlation for each trait-resolution pair, while italicized values represent the second-highest. Models incorporating a combination of citizen science and vegetation survey data (COMB) had the strongest correspondence for all traits and all resolutions, and models incorporating citizen science data only (CIT) exhibited the second-strongest correlations for most trait-resolution pairs. SCI models marginally outperformed COMB models on average but demonstrated consistently lower mean coefficients of variation and are therefore not shown here (Fig. 4a, b and Table S2). All trait values were transformed using a Yeo-Johnson power transform with the same parameters determined in the original transformation of the sPlot CWMs.

variance, lower prediction error, and improved accuracy in regions where citizen science observations substantially exceeded vegetation survey coverage. Given the sensitivity of these ecosystems to climate change, permafrost dynamics, treeline shifts, and associated greening

and browning trends, improving the spatial robustness of trait models is increasingly important[42].

The benefits of integrating multiple data sources were most pronounced at finer spatial resolutions (e.g., 1 km²), where environmental conditions are more homogeneous and local-scale species assemblages are better captured. At these scales, the broader species occurrence data improved spatial coverage, expanding the area of applicability and enabling models to generalize to a wider range of environments. However, at coarser resolutions, the distinction between SCI and COMB models diminished, as spatial aggregation reduced variance in the predictor and trait spaces.

Model accuracy, however, followed a slightly different pattern (Fig. 5d). Some traits—such as leaf area, seed mass, and stem density—exhibited higher predictive accuracy at coarser resolutions, while others plateaued or declined beyond 55 km² resolution. This trend suggests that certain traits may align more strongly with large-scale climatic gradients, while others are driven by localized environmental and ecological factors, as well as land use and local management. These findings emphasize the need for trait-specific assessments of spatial scaling effects to refine predictions across different functional trait dimensions.

Trait maps generated using the combined data approach (COMB) exhibited higher accuracy compared to previous global trait mapping efforts when validated against independent community-weighted mean (CWM) trait data (Table 2). This improvement is likely due to the complementary nature of the input data sources, allowing COMB to leverage the strengths of both vegetation surveys and citizen science observations while mitigating their individual limitations. Notably, models trained exclusively on citizen science data (CIT), which never incorporated vegetation survey data, also outperformed most previously published trait maps. These findings support those of Wolf et al.[28], reinforcing the value of crowdsourced species observations in trait mapping. Despite the biases inherent in opportunistic citizen science data, the sheer volume and geographic coverage of these datasets appear to encode meaningful ecological patterns.

These improvements also likely reflect differing trait representation and underlying methodological assumptions and goals. For example, several previous trait mapping efforts have relied primarily on remote sensing proxies or broad-scale climate-trait correlations, which effectively capture dominant vegetation patterns but may not fully represent the trait variation present across plant communities. A key distinction of our approach is its ability to aggregate species-level trait data from multiple sources, allowing for a direct estimation of community trait composition rather than relying on aggregated plant functional type (PFT) classifications. Trait products that integrate discrete land cover or PFT information—particularly those derived from top-of-canopy remote sensing—have been shown to better reflect the traits of the most dominant functional groups within a given region rather than CWM traits[27]. Here, it was our aim to capture traits across the full vertical structure of plant communities, a perspective particularly relevant for applications in ecosystem process modeling, functional diversity assessments, and dynamic vegetation models[1,12,16].

To model each trait, 150 Earth observation and environmental predictors representing approximately 19 billion observations were utilized, prompting critical consideration of whether such a highly data-intensive approach is justified. Given the scale of these inputs, we calculated predictor importance using feature permutation to quantify the relative influence of a given predictor or predictor set on prediction quality (Fig. S5).

MODIS surface reflectance was, on average, the most influential predictor set overall, followed by bioclimatic variables and soil properties, while canopy height and vegetation optical depth played a smaller role. These results align with prior research demonstrating that many functional traits are primarily structured along climate and, to a lesser extent, soil gradients, while their fine-scale variability may be

better captured through optical remote sensing[1,2,43,44]. Of course, such trends depend heavily on scale and the trait in question, and individual trait models exhibited distinct responses to different predictor types (Fig. S6). Stem conduit density, for example, was most strongly influenced by a WorldClim bioclimatic variable, specific leaf area by MODIS reflectance, and rooting depth by SoilGrids. Further, canopy height and vegetation optical depth unsurprisingly contributed most to modeling structural traits like plant height and leaf width. These results suggest that no single environmental predictor is universally superior at predicting all plant functional traits and affirm that varied predictor data may be necessary to begin to encode the complexity of the biosphere and its functional diversity.

Despite the advantages of integrating multiple crowdsourced datasets, sampling biases remain a challenge. Vegetation surveys, citizen science data, and trait databases each exhibit geographic clustering and taxonomic gaps due to accessibility constraints, observer behavior, and historical research focus in the global north[38,39,45]. In African savannas, for example, grasses—despite being dominant across vast regions—are underrepresented in citizen science observations due to the difficulty of taking meaningful photographs of them, specialized identification requirements, and morphological similarities that complicate species-level classification. Likewise, vegetation surveys often target specific communities of interest rather than providing representative coverage at broader spatial scales. Addressing these biases will require targeted efforts to improve taxonomic and seasonal coverage in trait-mapping initiatives, and the rapid expansion of citizen science projects and increased collaboration among scientists present valuable opportunities to close these gaps in the near future[28,29].

The influence of sampling density on predicted trait values is particularly evident in regions with sharp differences in observation coverage. In Portugal, for example, where CIT observation density is high, but SCI surveys are sparse, trait patterns differ noticeably from those in neighboring Spain, where fewer CIT observations are available. This discrepancy likely stems from the inclusion of multiple country-wide forest inventories from Portugal in the GBIF database, which may introduce a taxonomic bias in favor of woody plants. The resulting contrast in sampling density and methodology across the Iberian Peninsula's unique environmental conditions may, in turn, confound model prediction, as indicated by the high coefficients of variation observed within Portugal compared to the relatively low variance in Spain and western France. These findings highlight the need for systematic bias corrections[41], such as spatial filtering or weighting schemes, to reduce observation-driven artifacts in trait predictions while also emphasizing the benefits of more comprehensive vegetation surveys.

Expanding citizen science and vegetation survey datasets enhances our understanding of plant community composition, but trait mapping remains constrained by the limited availability of in situ trait measurements. Most species observed in this study lacked corresponding trait records in the TRY database, with only 28% of citizen science observations and 43% of vegetation survey species records successfully matched (though ~85% of all observations still fell within these species). This "Hutchinsonian shortfall" is by no means limited to this study, however, and despite ongoing database expansion and advanced gap-filling techniques, additional strategies are needed to bridge these gaps[25,46,47]. Computer vision, for instance, presents a promising opportunity to leverage the vast number of citizen science photographs to infer certain missing traits[48], not only improving trait coverage but also complementing traditional trait measurement efforts.

Even when species observations are successfully matched to trait databases, the "naive" approach of assigning species-mean trait values overlooks intraspecific trait variation (hereafter referred to as "intraspecific variation") across environmental gradients. While intraspecific variation may contribute less to trait patterns at large spatial scales, it remains a key factor in functional diversity assessments[24,49–51]. Importantly, the relevance of intraspecific variation varies by trait; for example, plant height exhibits substantial within-species variability, whereas specific leaf area (SLA) tends to be more stable[52]. Approaches such as incorporating in situ trait measurements or local PFT conditions within a given spatial radius may help capture intraspecific variation, though they often limit the number of species included per grid cell[44,53]. Future trait-mapping efforts should explore integrating intraspecific variation-aware modeling approaches to improve ecological realism in global trait predictions.

The integration of diverse data sources in trait mapping provides a powerful framework for capturing plant functional diversity. While our approach emphasizes methodological innovation, these advances enable critical ecological applications, such as directly addressing data gaps in trait-based biome classification[54] and providing the spatially continuous, high-resolution trait maps essential for next-generation process-based vegetation models[55]. As ecological research increasingly demands more traits, higher accuracy, and finer spatial resolution to understand ecosystem responses to global change, such methodological developments become foundational to expanding the horizons of trait-based ecology. Nevertheless, several avenues remain for advancing these methodologies. One major advantage of our approach is its adaptability; beyond linking species observations to mean trait values, it allows for the estimation of trait distributions, including quantiles, ranges, and variance. This flexibility could also be leveraged to better represent the variability within specific plant functional types (PFTs), particularly for groups such as grasses and shrubs, where within-PFT variation is often masked by broad community-level averaging[27,56]. Additionally, multi-label inference, an approach that enables models to predict multiple correlated traits simultaneously, could enhance predictive accuracy by leveraging known relationships among traits[57–60]. More extensive refinement of trait-specific predictor selection could also improve model efficiency by emphasizing the most relevant environmental variables for each trait. Trait predictions might be further improved by constraining species observations to those characteristic of specific vegetation formations, or PFTs, to ensure that trait-environment relationships are inferred within ecologically coherent units[2]. Beyond static trait predictions, future research could also focus on modeling key functional diversity (FD) and biodiversity metrics instead of solely community-weighted mean (CWM) traits. Understanding the global distribution of FD at high resolution may shed more light on the debated role of FD in ecosystem resilience and resistance to environmental change[30,61–63].

Future advancements in remote sensing technologies offer significant opportunities for refining trait models. The increasing availability and planned expansion of hyperspectral imaging, satellite-derived canopy fluorescence, and high-resolution structural metrics will provide new avenues for detecting fine-scale trait variability[64–66]. Moreover, incorporating temporally resolved environmental predictors—such as ECMWF Reanalysis in combination with the high temporal resolution of MODIS and other remote sensing missions—could enable the development of multi-temporal trait maps, providing dynamic snapshots of how trait distributions respond to environmental fluctuations[67,68]. By leveraging these time-series data, future trait models could better capture phenological shifts, disturbance responses, and other temporal dynamics that influence plant functional traits across different ecosystems. However, while environmental predictors may include high temporal resolution, the more restricted temporal character of trait measurements will likely remain a limiting factor.

This study presents a generalizable framework that integrates crowdsourced biodiversity data with high-resolution Earth observation to tackle longstanding challenges of extrapolation due to data sparseness in global plant trait mapping. By combining expert

vegetation surveys, citizen science-driven biodiversity observations, and trait measurement databases, our approach maximizes spatial coverage, trait representation, and predictive accuracy while mitigating individual data source limitations. However, these advancements were only made possible through the collective efforts of the scientific community and the broader public, facilitated by the growing framework of open, accessible, and reusable data. This collaborative, data-driven paradigm represents a fundamental shift in how we approach long-standing challenges in Earth system science. As curated expert data collections as well as crowdsourced biodiversity and trait databases continue to expand and data-sharing initiatives grow, future work should focus on improving bias correction, capturing trait variability, and integrating temporal dynamics to further refine functional trait predictions. By fostering continued collaboration between researchers, institutions, and citizen scientists, we can move toward a more comprehensive and globally representative understanding of plant functional diversity.

## Methods

### Trait data from the TRY plant trait database
Data for 31 plant functional traits linked to key ecological processes (Table 1) and measured across 74,245 species were retrieved from the gap-filled TRY Plant Trait Database, version 5 (www.try-db.org)[14,25,69–72]. The TRY database is an aggregation of trait measurements from individual plants, often with multiple trait measurements per species. Despite providing robust species mean trait values, the collection remains sparse and heterogeneous across environments and plant life stages in some areas, and so gap-filling utilizing Bayesian hierarchical probabilistic matrix factorization was implemented by the maintainers to expand its coverage[47,69]. Given the wide variation in functional trait distributions, trait data were transformed using the Yeo–Johnson power transformation to standardize the distributions prior to model training[73]. Power transformation was performed using the sklearn Python package (v1.4.2).

### Citizen science plant observations
Vascular plant species observations were retrieved from the Global Biodiversity Information Facility (GBIF; www.gbif.org) on 10 April 2024. Initial download parameters selected only observations that were members of Tracheophyta, included a geospatial reference, were marked as "present", and were non-cultivated. Prior to filtering, the GBIF data included 339,971,350 observations of 314,217 plant species from 12,645 datasets across 113 countries[74]. Included in the GBIF data were nearly 100 million observations from popular citizen science initiatives, including more than 31 million observations from the iNaturalist Research-grade Observations dataset and over 24 million observations from Observation.org and Pl@ntNet combined[32,33,75]. Importantly, GBIF aggregates observations from more than just popular citizen science applications, but given their predominant presence in the database as well as the large variability among the other data sources, we chose to refer to the totality of GBIF as "citizen science" for ease of readability and accessibility. Before spatial aggregation (see details below), GBIF observations were randomly subsampled at each spatial resolution so that each resulting grid cell contained no fewer than 10 and no more than 500 observations. The lower threshold of 10 observations per cell was chosen as a heuristic compromise to ensure sufficient data density while maintaining broad spatial coverage. We acknowledge this trade-off and note that optimizing such thresholds remains an open area for future research.

### Vegetation survey data
Vegetation survey data were retrieved from sPlot, a collation of worldwide vegetation plots including plot-level species relative coverage[29]. sPlot version 4.0 includes 52,942,365 occurrences of 91,603 species across 2,534,183 plots in 147 countries. Though GBIF

observations cover a much larger geographic extent (Fig. 1), sPlot surveys have the advantage of incorporating a more strategic sampling design, providing true absence information, and documenting species relative cover. Therefore, after matching with traits from TRY, sPlot can provide community-level trait statistics, such as community-weighted mean, median, and percentiles. This positions sPlot as a preferred alternative to GBIF in areas of overlapping or missing spatial coverage, as well as a benchmark against which trait extrapolation products can be compared.

### Assigning trait values to species observations and surveys
Following the methodology described by Wolf et al.[28], we calculated species-level means for the selected traits and assigned them to all matching species observations in both GBIF and sPlot using species name as the key matching criteria. PFT were not used in this assignment process, and were only considered when training models on observations from specific PFT combinations (not used in this study). Although a simple species-level mean does not capture intraspecific variation, it avoids the additional biases introduced by the spatially uneven availability of geo-referenced trait data and remains a practical strategy given current global data limitations; nonetheless, we acknowledge that intraspecific variation plays a crucial role in trait ecology and should be incorporated where feasible in future work. Due to inconsistencies in species name formatting, all species identifiers in each dataset were first truncated to the first two words containing their primary binomial names and matched in a case-insensitive manner. Not all species present in GBIF and sPlot were present in TRY, and so some species were excluded (see "Results"), though similar matching efforts have shown that the species that remained still likely represent plant communities[29,76]. For sPlot observations, plot-level community-weighted mean trait values were calculated by multiplying the relative cover of each species by the mean trait from TRY and averaging the resulting weighted values for each plot.

### Spatial aggregation of derived trait values
In order to match the derived GBIF and sPlot trait values with gridded Earth observation data, we spatially aggregated both trait subsets. GBIF observation locations and sPlot plot locations were first projected from geographic coordinates into Equal-Area Scalable Earth (EASE) Grids, version 2.0, Global Cylindrical (EPSG: 6933). The GBIF individual trait values and sPlot plot-level community-weighted mean (CWM) trait values were each aggregated into separate raster grids at five spatial resolutions: 1, 22, 55, 111, and 222 $km^2$ using the mean. Because different plant traits exhibit spatial variation at different scales, with some traits responding primarily to local environmental conditions while others are driven by broad-scale climatic gradients[77,78], we generated trait training data at multiple resolutions to optimize model performance and assess scale-dependent ecological patterns. This approach enables identification of the optimal spatial scale for different functional traits and supports diverse research applications ranging from fine-scale ecosystem modeling to broad-scale biogeographic analyses. As a result, the GBIF grids represented frequency-weighted mean trait values, which have been shown to approximate CWM traits[28], while the sPlot grids contained mean CWM values. Correlations between the sparse sPlot and GBIF grids are presented in Table S3. For the combined trait data subset (COMB), GBIF and sPlot aggregations were merged, with sPlot values taking precedence in grid cells where trait means from both datasets were present. All spatial manipulation was performed using the Python libraries of pandas (v2.2.2), dask (v2024.9.0), xarray (v2024.9.0), and rasterio (v1.4.1).

### Gridded Earth observation data
Climate data have been shown to largely drive the global variation in plant traits, and previous studies have successfully predicted plant

traits via the environment[44,53,79–82]. More specifically, temperature influences photosynthesis by integrating energy availability from solar radiation. Plant productivity, reliant on photosynthesis, is also tied to water availability, making precipitation dynamics a crucial indicator of plant functional configuration[83]. Bioclimatic variables were extracted from the WorldClim database with a resolution of $0° 10'0''$[22]. The following bioclimatic variables were selected: annual mean temperature (BIO1), temperature seasonality (BIO4), temperature annual range (BIO7), annual precipitation (BIO12), precipitation of wettest month (BIO13), precipitation of driest month (BIO14), and precipitation seasonality (BIO15). To calculate precipitation annual range (BIO13–14) we subtracted precipitation of driest month (BIO14) from precipitation of wettest month (BIO13) and later discarded them, using only BIO13–14 to explain precipitation variability. As mentioned above, BIO1 and BIO12 are known predictors of plant traits. Likewise, climate variables associated with range and seasonality have also been shown to correlate with plant traits. Therefore, we specifically chose the additional four climate variables (BIO4, BIO7, BIO13–14, BIO15) to capture the annual variations in the site-specific climatic conditions.

Given that plant canopies are configured to interact with light, their optical reflectance properties can inform their functional trait configuration[44,84]. When considering the prospect of perpetually updated plant trait models, remote sensing data is especially appealing given its high spatial and temporal availability. For this study, surface reflectances in the visible and infrared spectra were collected from the MODIS MOD09GA v061 (MODIS/Terra Surface Reflectance Daily L2G Global 1 km and 500 m SIN Grid) dataset[19]. MODIS was selected due to its more reliable and continuous large-scale coverage when compared to other optical remote sensing missions like Landsat and Sentinel-2. Bands 1–5 (620–670 nm, 841–876 nm, 459–479 nm, 545–565 nm, and 1230–1250 nm, respectively) were used, as well as an NDVI band, which was calculated using bands 1 (red) and 2 (near-infrared). In order to preserve as much spatial and temporal information as possible, surface reflectances for each band were obtained at $1 km^2$ resolution and temporally aggregated into 12 monthly averages spanning 20 years from March 2000 through March 2020, yielding 72 total predictors ($12 \times 5$). These operations were performed using Google Earth Engine.

Because soil properties influence nutrient and water availability–key factors in plant resource acquisition and trait composition–we included high-resolution soil data under the assumption that it could help explain some of the trait variance[47]. Mean soil properties were obtained from the SoilGrids 2.0 dataset–global soil property predictions at six standard depth intervals at a spatial resolution of 250 meters–produced and hosted by the International Soil Reference and Information Centre (ISRIC)[21]. All available soil properties at all depth intervals were used, including pH, soil organic carbon content, bulk density, coarse fragments content, sand content, silt content, clay content, cation exchange capacity, total nitrogen, soil organic carbon density, and soil organic carbon stock.

It is important to note that the predictor datasets used during model training for SoilGrids predictions include MODIS optical observations (middle and near-infrared bands) as well as WorldClim 2.0 bioclimatic variables, and therefore, some collinearity may exist between the predictors used in this study.

Vegetation optical depth (VOD) measures the opacity of vegetation to microwave radiation, indicating the amount and water content of plant biomass across landscapes. In addition to plant water content, VOD can inform ecosystem properties relevant to plant traits such as vegetation density, biomass, and environmental conditions[85,86]. Further, VOD can discriminate between vegetation types, a reliable predictor of plant traits[2,27,44,53,87]. Global, multi-sensor measurements of VOD have been made available as part of the global long-term microwave Vegetation Optical Depth Climate Archive (VODCA)[20]. We obtained all available measurements in the C, Ku, and X bands from all available years (C-band: 2002–2018; Ku-band: 1987–2017; X-band:

1997–2018) at a native spatial resolution of 0.25° and upsampled all measurements to $1 km^2$ using bilinear interpolation. Bilinear interpolation was utilized over nearest-neighbor upsampling (NN) as NN tended to result in the imprinting of the 0.25° grid cells during model extrapolation. Three temporal aggregations were then computed across each band time series: mean and the 5th and 95th percentiles.

Canopy height not only has an obvious direct relationship with community-level plant height, but has also been shown to be correlated with above-ground traits like stem diameter, stem specific density, and seed mass, as well as below-ground and other hydraulic traits such as stem conductivity and potential root length and mass[57,88–90]. Additionally, taller plants directly affect the competitive landscape by reducing light availability to below-canopy communities as well as demanding more water and nutrients than smaller individuals due to increased transpiration and photosynthesis[91,92]. To represent global canopy height, we utilized the global canopy height and canopy height standard deviation products developed by Lang et al.[37] due to their uniquely high spatial resolution compared to other canopy height models.

All predictor datasets, with the exception of VOD, were obtained at resolutions similar to or finer than $1 km^2$, and so each dataset was first reprojected to EASE (EPSG: 6933) for standardized, area-consistent global referencing and then spatially resampled using weighted mean resampling to $1 km^2$ (~0.01°), $22 km^2$ (~0.2°), $55 km^2$ (~0.5°), $111 km^2$ (~1°), and $222 km^2$ (~2°) grids. As mentioned above, upsampling was required for VOD observations in order to be matched with other predictors and trait grids at $1 km^2$ and $22 km^2$ resolutions. Lastly, water and urban land cover pixels were masked from all predictor maps using ESA WorldCover 10m v100[93].

## Model training and validation

Prior to model training, environmental predictors and GBIF and sPlot trait data (labels) were matched by pixel coordinates and converted to tabular format. Though our machine-learning approach could tolerate missing values, pixels containing fewer than 60% of the predictors were excluded from the final training sets. Overall, the three subsets of trait data were matched with environmental predictors: vegetation survey trait community-weighted means (CWM) only (SCI), citizen science trait means only (CIT), and a combination of both, where vegetation survey CWMs were preferred when both were present (COMB; Table 3). This approach was repeated to generate training data for models at each resolution.

After matching by pixel coordinates, models were trained for all traits across each trait data subset using LightGBM gradient-boosting decision trees (Python library v4.5.0)[94]. LightGBM utilizes a suite of boosted decision trees and is capable of capturing complex, nonlinear relationships among high-dimensional tabular data while avoiding overfitting and remaining resource-efficient[35]. Additionally, LightGBM is capable of reasonably handling missing values by ignoring them during splitting, then allocating them to whichever side reduces loss the most. This was necessary as not all predictors had complete spatial overlap, and so, while observations that had more than 40% missing predictor values were filtered prior to training, some missing predictor values persisted depending on the spatial coverage of each particular trait.

When validating machine learning-based geospatial models, spatial autocorrelation must be considered[95,96]. Traditional validation methods such as random K-fold or leave-one-out cross-validation (CV) tend to produce overly optimistic results because they do not account for spatial clustering in training data or discrepancies between training and extrapolation feature spaces[95,97]. While probability sampling and design-based inference may be preferable for reference datasets with sufficiently broad and evenly-distributed spatial coverage, spatial K-fold cross-validation (hereafter referred to as "spatial cross-validation") has been shown to be a robust alternative for sparse, spatially

**Table 3 | Trait data subsets used for model training**

| Trait data subset | Source | Trait aggregation type | Training data size after spatial aggregation |
|---|---|---|---|
| Vegetation surveys (SCI) | sPlot | Community-weighted mean | 430,213 |
| Citizen-science observations (CIT) | GBIF | Frequency-weighted mean | 2,392,987 |
| Combined (COMB) | sPlot + GBIF | Community-weighted mean (preferred) + frequency-weighted mean | 2,646,876 |

Models for all traits were trained for each trait data subset at multiple resolutions and evaluated against held-out, spatially independent community-weighted mean traits using spatial K-fold cross-validation. In the case of the COMB trait data subset, SCI and CIT were merged with a preference for SCI values when both SCI and CIT data were present.

clustered datasets[98]. Because spatial cross-validation partitions data into spatially independent folds and evaluates model performance across distinct geographic regions, it provides a geographically explicit and transferability-aware validation framework. When combined with supplementary quality indicators—such as the coefficient of variation and area of applicability (AOA), discussed below—spatial cross-validation provides a more reliable assessment of model generalizability[36,95,99].

To construct spatial cross-validation folds, we first determined the spatial autocorrelation range of sPlot community-weighted mean (CWM) trait values by computing spherical semivariograms at a 1 km$^2$ resolution using the pykrige Python library (v1.7.2). We then overlaid a hexagonal grid with cells sized according to this range and randomly assigned each cell to one of five folds (Fig. S1). All trait observations from both sPlot and GBIF were assigned a fold ID based on their respective hexagons. To ensure balanced fold distributions, we ran 100 simulations of random fold assignments and selected the final configuration based on minimizing dissimilarity between folds. Dissimilarity was quantified using Kolmogorov-Smirnov tests, with the chosen fold assignment maximizing the mean $p$-value, thereby minimizing the likelihood of systematic differences between folds.

Model training and validation were performed using the leave-one-out (LOO) method, in which predictor-trait pairs from a single fold were set aside for model validation while fitting was performed on the remaining data, and the process was repeated for each remaining fold. All trait model variants (SCI, CIT, and COMB) were validated using vegetation survey community-weighted mean trait values from the held-out fold to ensure validation integrity and spatial independence of the validation data. Importantly, spatial cross-validation was not used for model selection or hyperparameter tuning; all model configurations were fixed prior to validation. As such, and consistent with best practices in the machine learning literature, a separate test set was not required[100,101].

Due to the significant imbalance in sample size between GBIF and sPlot pixels after spatial aggregation, samples in the combined trait data subset (COMB) were weighted based on their origin. sPlot vegetation surveys, which represent community-weighted means, were deemed more reliable than citizen science observations. Consequently, weights were assigned to GBIF observations inversely to their proportion relative to sPlot observations in the combined training set. Additionally, feature pruning was performed during model training based on automated sensitivity analysis via permutation feature importance. Features were dynamically removed during model ensemble creation if their removal improved model performance. Because final models were actually an ensemble of child models, not all child models necessarily utilized the same pruned feature set. Finally, models used to produce final trait prediction maps were trained on all available environmental predictor data.

The retrieval of predictor importance, or the weight given to a particular feature during model training, is an important aspect of model interpretability. Feature and dataset importance were calculated using the permutation importance method, in which the values of each feature or set of features (dataset) are randomly shuffled, and

model performance is measured[102]. The permutation importance is quantified as the performance difference when prediction is performed on data containing the shuffled feature. Because we implemented spatial cross-validation, feature importance was defined as the average feature importance across all CV folds.

## Model performance and spatial transferability

Citizen science trait values, given their opportunistic provenance, are noisy and biased[39]. As mentioned above, we validated all model runs against spatially independent sPlot community-weighted mean trait values that were not used during model training for each of the three trait data subsets (CIT, SCI, and COMB). Model performance was assessed using average normalized root mean squared error (nRMSE) and Pearson's correlation coefficient $r$ based on the combined observed versus predicted values of all spatial cross-validation runs. Following the standard practice in global trait mapping studies, we used Pearson's correlation coefficient as the primary metric for comparison rather than $R^2$[27,28]. To further assess model robustness, we calculated the pixel-wise coefficient of variation (COV) by generating predictions from each cross-validation model, stacking the results, and computing the COV for each pixel along the vertical stack as the ratio of the standard deviation to the mean prediction. All the above statistics were also assessed across biomes. Biomes were first retrieved from the Terrestrial Ecoregions of the World map and then consolidated from 14 biomes into 7 biome categories[103]. The biome-to-biome-category mapping can be found in the supplementary information (Table S4).

While spatial cross-validation is useful for minimizing spatial dependence on model error estimation, it may not be sufficient for explaining the transferability of model extrapolations in areas where predictor data vary significantly from the reference data[40,97,104]. In particular, we were interested in assessing whether the abundance and coverage of citizen science observations impart greater spatial transferability to citizen science-based models (COMB) compared to using less-abundant vegetation surveys alone. To address this, we applied the methods developed by Meyer & Pebesma[36] for determining the dissimilarity index (DI) between predictor data used during model training ("train") and unseen predictor data used for extrapolation ("new"), and thus the area of applicability (AOA) of the final trait maps. Briefly, dissimilarity in the predictor space was calculated by first computing the average minimum distance between observations in each cross-validation fold in "train" (i.e., the minimum distances between points in one fold from points in all other folds), followed by calculating the minimum distances between observations in the "new" data from the training data. The DI was then determined as "new" distances divided by the mean "train" distance. In line with existing literature, we defined the DI threshold as $1.5 \times IQR_{DI} + 75th \ percentile_{DI}$ of the cross-validated training data, with values below the threshold being within the AOA and values above the threshold (outliers) being outside it. DI and AOA were computed for all models at all resolutions[36]. To quantify whether the inclusion of GBIF data increased the AOA of trait models, we compared the AOA of all models across all training sets and calculated the change in overall extent.

It should be noted that the methods described by Meyer & Pebesma[36] do not address the handling of missing values. Because LightGBM can tolerate missing values and we did not do any data imputation prior to training, we had several options when calculating AOA: a) drop all observations in "new" and "train" that contained any missing features (and therefore misrepresent the actual spatial coverage of the training data as well as reduce feature space variance present in DI calculation compared to actual reference data used in training), likely resulting in an overly pessimistic AOA; b) drop features in both the "new" and "train" data that contained any missing values, likely resulting in an overly optimistic AOA as feature space complexity is reduced; or c) a "middle ground" approach of imputing missing values and assuming that the resulting predictor space is still a robust representation of the true reference data. To retain as much of the original signature of the true reference data when calculating dissimilarity, as well as to ensure that final AOA maps matched the geographic extent of the predictions, we chose option "c". However, we recognize that, given the novelty of this method, there is likely room for improvement. Missing value imputation was performed using the `NaNImputer` method from the Python "verstack" library (v4.1.4), which fits LightGBM regression models for each feature to fill missing values[105].

We generated continuous trait maps for all traits at all resolutions for the SCI and COMB trait subsets as stacked rasters containing the inference data, COV, and AOA mask. The rasters were written as cloud-optimized GeoTIFFs in EPSG:6933, and the inference and COV bands were quantized to signed 16-bit integer format.

To compare our trait maps with existing products, we used sPlot gridded CWMs as a benchmark, an established method for comparing global trait maps[27,28]. Because SCI and COMB models were trained using sPlot data—a distinguishing feature of this study—we used spatial cross-validation to prevent overlap and spatial autocorrelation between training and validation datasets.

Trait maps from nine existing products covering three foliar traits-specific leaf area, leaf nitrogen per area (leaf N [area]), and leaf nitrogen per mass (leaf N [mass])-were selected and spatially aligned with sPlot CWMs at the nearest common resolution (1, 22, 55, 111, or 222 km$^2$). Most products were originally created at a single native resolution, but to enable performance comparisons across scales, they were downsampled to coarser resolutions when necessary. Upsampling to finer resolutions was avoided to prevent unnecessary data manipulation. The final trait values were transformed using a Yeo−Johnson power transform with the lambda values determined in the original transformation of the sPlot gridded CWMs. Pearson's correlation coefficient $r$ between the transformed trait products and matched sPlot CWMs was then used to describe agreement. The specific foliar traits were selected due to their general commonality between the selected previous studies.

### Reporting summary
Further information on research design is available in the Nature Portfolio Reporting Summary linked to this article.

## Data availability
The Earth observation data used in this study are publicly available through Google Earth Engine (https://earthengine.google.com) or the Google Earth Engine Community Catalog (https://gee-community-catalog.org/). GBIF species occurrence data are available via the dataset citation provided in the ref. 75. The TRY gap filled trait data used in this study are available under restricted access due to the data sharing policies of contributing datasets within TRY. Access can be requested through the TRY Plant Trait Database (https://www.try-db.org) following their standard data request procedure. The sPlot vegetation survey data used in this study are available under restricted access to protect the interests of data contributors. Access can be requested by contacting the sPlot consortium through the German Centre for Integrative Biodiversity Research (iDiv) via G.D. or through the sPlot website (https://www.idiv.de/splot). After request processing, data are provided under a data use agreement. The global trait maps generated in this study have been deposited in Zenodo[106]. An interactive map viewer is available at https://global-traits.projects.earthengine.app/view/global-traits, and additional study resources can be found at https://planttraits.earth. Users of these maps should consult the coefficient of variation and area of applicability layers, as well as the model performance metrics provided in the raster metadata and Table S2, noting that performance varies across traits and biomes (Figs. 4b, 5c, S3 and Table S4). Source data underlying the figures are available at https://doi.org/10.5281/zenodo.18108765.

## Code availability
The code used to process data, train models, and generate trait maps in this study is available at https://github.com/dluks/cit-sci-trait-maps and archived on Zenodo at https://doi.org/10.5281/zenodo.18269445.

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

## Acknowledgements

This study was funded by the German Research Foundation (DFG) within the framework of BigPlantSens (Assessing the Synergies of Big Data and Deep Learning for the Remote Sensing of Plant Species; project no. 444524904) and PANOPS (Revealing Earth's plant functional diversity with citizen science; project no. 504978936). D.L. and T.K. thank the European Space Agency for funding the "FORTRACK" project via the ESA CLIMATE SPACE: Climate-Biodiversity studies. The study is supported by the TRY initiative on plant traits (http://www.try-db.org) and the sPlot consortium (http://www.idiv.de/splot). The TRY initiative and database are hosted, developed, and maintained by J.K. and G. Boenisch (Max Planck Institute for Biogeochemistry, Jena, Germany), currently supported by Future Earth/bioDISCOVERY and the German Centre for Integrative Biodiversity Research Halle-Jena-Leipzig (iDiv, DFG-FZT 118, 202548816). sPlot is a strategic project of iDiv and is supported by the German Research Foundation (DFG-FZT 118, 202548816). F.M.S. gratefully acknowledges the support of the Italian Ministry of University and Research, under the Maria Levi Montalcini programme. S.W. was funded by the German National Research Data Infrastructure for Biodiversity, NFDI4Biodiversity, a DFG project, project no. 442032008 and by the European Space Agency Climate Change Initiative (ESA-CCI) Tipping Elements SIRENE project (contract no. 4000146954/24/I-LR). CFD acknowledges funding by the German Research Foundation (DFG) through CRC 1537 Ecosense and EXC 3127 Future Forests. F.G.'s surveying efforts were funded by the Swiss National Science Foundation Postdoctoral Fellowships (TMPFP2_217531). F.R. acknowledges the support of Coordenação de Aperfeiçoamento de Pessoal de Nível Superior (CAPES) for postdoctoral fellowships (N˚88887.974741/2024-00). J.A., J. Dolezal, and K. Korznikov were supported by the research grant 25-15727S of the Czech Science Foundation and long-term research development project No. RVO 67985939 of the Institute of Botany of the Czech Academy of Sciences. This work would not have been possible without the contributions of ecologists and vegetation surveyors who diligently sampled field plots, curated datasets, and shared them through accessible databases. We also acknowledge the vital role of citizen scientists engaged in platforms such as iNaturalist and Pl@ntNet, whose time, observations, and local knowledge have been crucial in assembling high-quality, research-ready data.

## Author contributions

D.L. and T.K. conceived the study. D.L. designed the methodology and led the analysis. S.W., D.S., and K.G. contributed to the development of analytical tools and statistical modeling. Data collection and processing were carried out by C.V., D.H., G.J.A.H., S.T., S.P., F.G., H.K., M.S., H.C., B.G., J. Dolezal, R.P., A.G., C.D., F.N., J.W., A.L.G., M.J.M., M.C., J.L., D.T., J. Dengler, S.Ś., J.A., L.M., A.N.N., K. Kakinuma, P.R., Z.S., R.T., M.Z.H., F.R., J.H., M.C.M.M., J.K.M., M.A.E., and K. Korznikov, who also provided critical feedback on data quality and interpretation. Manuscript writing was led by D.L., with substantial contributions from T.K., S.W., D.S., C.F.D., J.K., H.B., F.M.S., G.D., and A.M.M., and all authors reviewed and edited the manuscript.

## Funding

## Competing interests

The authors declare no competing interests.

## Additional information

**Daniel Lusk** [1]✉, **Sophie Wolf** [2], **Daria Svidzinska**[2], **Carsten F. Dormann** [3], **Jens Kattge** [4,5], **Helge Bruelheide** [4,6], **Francesco Maria Sabatini** [7], **Gabriella Damasceno** [4,6], **Álvaro Moreno Martínez** [8], **Cyrille Violle** [9], **Daniel Hending** [10], **Georg J. A. Hähn** [7], **Solana Tabeni** [11], **Shyam Phartyal** [12], **Fernando Gonçalves** [13], **Holger Kreft** [14], **Marco Schmidt** [15], **Han Chen** [16,17], **Behlül Güler**[18], **Jiri Dolezal** [19,20], **Remigiusz Pielech**[21], **Anaclara Guido**[22], **Ciara Dwyer** [23], **Francesca Napoleone** [24], **Jacob Willie** [25], **André Luís Gasper** [26], **Manuel J. Macía** [27,28], **Milan Chytry** [29], **Jonathan Lenoir** [30], **Dinesh Thakur** [19], **Jürgen Dengler** [31], **Sebastian Świerszcz**[32], **Jan Altman** [19,33], **Ladislav Mucina** [34,35], **Ashish N. Nerlekar**[36,37], **Kaoru Kakinuma** [38], **Pravin Rawat**[39], **Zvjezdana Stančić** [40], **Riccardo Testolin**[7], **Mohamed Z. Hatim** [41], **Flávio Rodrigues** [42],

**Jürgen Homeier** ⑩ [43], **Marcia C. M. Marques** ⑩ [44], **James K. McCarthy** ⑩ [45], **M. A. El-Sheikh** ⑩ [46], **Kirill Korznikov**[19], **Kilian Gerberding** ⑩ [1] **& Teja Kattenborn** ⑩ [1]

[1]Chair of Sensor-based Geoinformatics (geosense), University of Freiburg, Freiburg, Germany. [2]Remote Sensing Centre for Earth System Research, Leipzig University, Leipzig, Germany. [3]Department of Biometry and Environmental System Analysis, University of Freiburg, Freiburg, Germany. [4]German Centre for Integrative Biodiversity Research (iDiv) Halle-Jena-Leipzig, Leipzig, Germany. [5]Max Planck Institute for Biogeochemistry, Jena, Germany. [6]Institute of Biology/Geobotany and Botanical Garden, Martin Luther University Halle-Wittenberg, Halle, Germany. [7]BIOME Lab, Department of Biological, Geological and Environmental Sciences (BiGeA), Alma Mater Studiorum University of Bologna, Bologna, Italy. [8]Image Signal Processing Group, Image Processing Laboratory (IPL), University of Valencia, Paterna, Spain. [9]CEFE, CNRS, EPHE, IRD, University of Montpellier, Montpellier, France. [10]Department of Biology, University of Oxford, Oxford, UK. [11]Instituto Argentino de Investigaciones de las Zonas Áridas, CONICET, Mendoza, Argentina. [12]Department of Forestry, Mizoram University, Aizawl, India. [13]Department of Evolutionary Biology and Environmental Studies, University of Zurich, Zurich, Switzerland. [14]Biodiversity, Macroecology & Biogeography, University of Göttingen, Göttingen, Germany. [15]Palmengarten der Stadt Frankfurt am Main, Frankfurt am Main, Germany. [16]College of Grassland Science, Inner Mongolia Agricultural University, Hohhot, China. [17]Institute for Global Change Biology, and School for Environment and Sustainability, University of Michigan, Ann Arbor, MI, USA. [18]Biology Education, Dokuz Eylül University, Buca, Izmir, Turkey. [19]Institute of Botany of the Czech Academy of Sciences, Trebon, Czechia. [20]Faculty of Science, University of South Bohemia, České Budějovice, Czechia. [21]Institute of Botany, Faculty of Biology, Jagiellonian University, Kraków, Poland. [22]Instituto de Ecología y Ciencias Ambientales, Facultad de Ciencias, Universidad de la República, Montevideo, Uruguay. [23]Centre for Environmental and Climate Science, Lund University, Lund, Sweden. [24]Sapienza University of Rome, Rome, Italy. [25]Centre for Research and Conservation, Royal Zoological Society of Antwerp, Antwerp, Belgium. [26]Universidade Regional de Blumenau, Blumenau, Brazil. [27]Departamento de Biología (Botánica), Universidad Autónoma de Madrid, Madrid, Spain. [28]Centro de Investigación en Biodiversidad y Cambio Global (CIBC-UAM), Universidad Autónoma de Madrid, Madrid, Spain. [29]Department of Botany and Zoology, Faculty of Science, Masaryk University, Brno, Czechia. [30]UMR CNRS 7058 "Ecologie et Dynamique des Systèmes Anthropisés" (EDYSAN), Université de Picardie Jules Verne, Amiens, France. [31]Institute of Natural Resource Sciences (IUNR), Zurich University of Applied Sciences (ZHAW), Wädenswil, Switzerland. [32]Institute of Agroecology and Plant Production, Wrocław University of Environmental and Life Sciences, Wrocław, Poland. [33]Faculty of Forestry and Wood Sciences, Czech University of Life Sciences Prague, Prague, Czechia. [34]Iluka Chair in Vegetation Science and Biogeography, Harry Butler Institute, Murdoch University, Perth, WA, Australia. [35]Department of Geography and Environmental Studies, Stellenbosch University, Stellenbosch, South Africa. [36]Department of Plant Biology, Michigan State University, East Lansing, MI, USA. [37]Program in Ecology, Evolution, and Behavior, Michigan State University, East Lansing, MI, USA. [38]Korea Advanced Institute of Science and Technology, Daejeon, South Korea. [39]ICFRE- Himalayan Forest Research Institute, Shimla, Himachal Pradesh, India. [40]Faculty of Geotechnical Engineering, University of Zagreb, Zagreb, Croatia. [41]Plant Ecology and Nature Conservation Group, Environmental Sciences Department, Wageningen University and Research, Wageningen, The Netherlands. [42]Plant Ecology Laboratory (LABEV), UNEMAT, Nova Xavantina, MT, Brazil. [43]Faculty of Resource Management, HAWK University of Applied Sciences and Arts, Göttingen, Germany. [44]Departamento de Botânica, SCB, Universidade Federal do Paraná, Curitiba, Brazil. [45]Manaaki Whenua - Landcare Research, Lincoln, New Zealand. [46]Botany and Microbiology Department, College of Science, King Saud University, Riyadh, Saudi Arabia. ✉e-mail: daniel.lusk@geosense.uni-freiburg.de

