## [Transparent Peer Review File · Nature Communications]

Crowdsourced biodiversity monitoring fills gaps in global plant trait mapping

Corresponding Author: Mr Daniel Lusk

Version 0:

Reviewer comments:

Reviewer #2

(Remarks to the Author)

The paper presents a dataset including a collection of plant traits scaling up from plot level to grids at up to 1 km² resolution across the globe. The work will provide estimations and references for global-scale ecological studies and inputs for improving process-based modelling. Some of the traits have been recently mapped using satellite remote sensing data at the global scale, such as SLA, leaf nitrogen and phosphorus (Dechant et al. 2024), and some traits were estimated purely based on climate variables, such as by Jiaze Li & Iain Colin Prentice, 2024, which includes 16 functional traits. The novelty of this study lies in its comprehensive use of satellite data for 31 plant traits and comprehensive analysis of data uncertainties. It fills a gap for a consistent global scale mapping for so many traits. The presentation of the paper is clear, and the structure is well organized. The methodology and data analysis are scientifically sound and meet the expected standards in related field. The description of data processing methods is detailed and easy to follow. The online view of global trait maps is user friendly.

references

Dechant, B. et al. Intercomparison of global foliar trait maps reveals fundamental differences 878 and limitations of upscaling approaches. *Remote Sensing of Environment* 311, 114276. doi:10. 879 1016/j.rse.2024.114276 (2024)

Li, J. & Prentice, I. C. Global patterns of plant functional traits and their relationships to 922 climate. *Communications Biology* 7, 1–14. doi:10.1038/s42003-024-06777-3 (2024)

(Remarks on code availability)

Reviewer #3

(Remarks to the Author)

(Remarks on code availability)
